# Impact of Mass Workplace COVID-19 Rapid Testing on Health and Healthcare Resource Savings

**DOI:** 10.3390/ijerph18137129

**Published:** 2021-07-03

**Authors:** Francesc López Seguí, Jose Maria Navarrete Duran, Albert Tuldrà, Maria Sarquella, Boris Revollo, Josep Maria Llibre, Jordi Ara del Rey, Oriol Estrada Cuxart, Roger Paredes Deirós, Guillem Hernández Guillamet, Bonaventura Clotet Sala, Josep Vidal Alaball, Patricia Such Faro

**Affiliations:** 1Fight AIDS and Infectious Diseases Foundation, 08916 Badalona, Spain; atuldra@flsida.org (A.T.); msarquella@flsida.org (M.S.); brevollo@flsdia.org (B.R.); jmllibre@flsida.org (J.M.L.); rparedes@irsicaixa.es (R.P.D.); bclotet@irsicaixa.es (B.C.S.); 2North Metropolitan Primary Care Directorate, Catalan Institute of Health, 08916 Badalona, Spain; gterritorial.mn.ics@gencat.cat (J.A.d.R.); oestrada@gencat.cat (O.E.C.); 3Health Safety and Emergencies Unit SEAT CUPRA, the Companies of the Volkswagen Group in Spain, 08916 Badalona, Spain; jose.navarrete@seat.es; 4Central Catalonia Primary Care Directorate, Catalan Institute of Health, Sant Fruitos de Bages, 08272 Barcelona, Spain; guillemhg98@gmail.com (G.H.G.); jvidal.cc.ics@gencat.cat (J.V.A.); 5Health Promotion in Rural Areas Research Group, Gerencia Territorial de la Catalunya Central, Institut Catala de la Salut, Sant Fruitos de Bages, 08272 Barcelona, Spain; 6Unitat de Suport a la Recerca de la Catalunya Central, Fundacio Institut Universitari per a la Recerca a l’Atencio Primaria de Salut Jordi Gol i Gurina, Sant Fruitos de Bages, 08272 Barcelona, Spain

**Keywords:** workplace testing, economic analysis, COVID-19, asymptomatic screening, mass testing, employee population health, return to work practices, SARS-CoV-2, surveillance, workplace mitigation

## Abstract

***Background***: The epidemiological situation generated by COVID-19 has cast into sharp relief the delicate balance between public health priorities and the economy, with businesses obliged to toe the line between employee health and continued production. In an effort to detect as many cases as possible, isolate contacts, cut transmission chains, and limit the spread of the virus in the workplace, mass testing strategies have been implemented in both public health and industrial contexts to minimize the risk of disruption in activity. ***Objective***: To evaluate the economic impact of the mass workplace testing strategy as carried out by a large automotive company in Catalonia in terms of health and healthcare resource savings. ***Methodology***: Analysis of health costs and impacts based on the estimation of the mortality and morbidity avoided because of screening, and the resulting savings in healthcare costs. ***Results***: The economic impact of the mass workplace testing strategies (using both PCR and RAT tests) was approximately €10.44 per test performed or €5575.49 per positive detected; 38% of this figure corresponds to savings derived from better use of health resources (hospital beds, ICU beds, and follow-up of infected cases), while the remaining 62% corresponds to improved health rates due to the avoided morbidity and mortality. In scenarios with higher positivity rates and a greater impact of the infection on health and the use of health resources, these results could be up to ten times higher (€130.24 per test performed or €69,565.59 per positive detected). **Conclusion**: In the context of COVID-19, preventive actions carried out by the private sector to safeguard industrial production also have concomitant public benefits in the form of savings in healthcare costs. Thus, governmental bodies need to recognize the value of implementing such strategies in private settings and facilitate them through, for example, subsidies.

## 1. Introduction

Since the outbreak of the COVID-19 epidemic in February 2020, governments have been faced with the dilemma of limiting the spread of the disease by strict and long-term community lockdowns to limit transmission rates, while not causing serious or permanent damage to the economy [1,2,3,4,5]. A less drastic alternative to preventive confinement is carrying out comprehensive screening for the early detection and segregation of cases. This can greatly reduce transmission rates, particularly as presymptomatic and asymptomatic carriers of the virus may account for approximately 40% of all transmissions [5,6,7,8]. Such mass screening strategies have been put into practice in healthcare systems around the world [3,9,10,11,12,13,14,15], with the general consensus being that such screening must take place early and systematically, yield consistently reliable results, reach the maximum possible population, ensure the full isolation of positives, and involve the use of tracers to identify and alert close contacts.

While such workplace screening for COVID-19 has been carried out routinely since the start of the pandemic in health care contexts such as hospitals, there is clearly also a need for it in industrial contexts, where conditions often hinder physical distancing between individual workers [16,17,18,19,20,21,22,23,24,25,26]. The automobile manufacturer SEAT, S.A., a leader in the sector and the only Spanish auto-maker that designs, develops, produces, and markets its products entirely locally, employs 15,000 workers directly and up to 100,000 indirectly, at facilities located mostly in Catalonia within the greater Barcelona metropolitan area. Because SEAT is a key driving force for the regional economy, the potential socio-economic impact of any disruption to its production chain extends far beyond the company itself. For this reason, starting as early as February 2020, the company’s Occupational Risk Prevention Service began to apply prevention measures against COVID-19. A policy of mass screening through diagnostic testing for the disease was initiated in April of the same year, making SEAT a pioneer in the application of these techniques in the manufacturing sector.

The several existing studies that have sought to determine the impact of mass screening strategies have yielded heterogeneous results, showing various effects on health and savings in the use of health resources, depending on, among other factors, the positivity rate of the tested population [27,28,29]. All of these studies have adopted a social perspective, largely because the effects of the infection are indiscriminate in the general population and especially impact the public healthcare sector. However, it is important to examine the effectiveness of mass testing strategies in industry as well, as although the costs are borne by the company privately, part of the resulting benefits constitute positive externalities beyond the strictly business perspective, such as the savings in health resources that result from the interruption of transmission chains. In this context, the present paper takes the mass workplace COVID-19 screening experience at SEAT as a case study to quantify, using traditional cost analysis methods, the effect of such testing on public health and healthcare resource savings.

## 2. Methodology

### 2.1. Setting

The object of this study is the series of screenings of SEAT workers carried out by the company’s Health, Safety, and Emergencies Unit from 22 March 2020–24 March 2021, a period that included the moments of maximum incidence of the virus in Spain [30]. All of the workers in the production chain were tested twice a week. The sample analyzed thus comprises the results of 188,552 COVID-19 diagnostic tests. During this time, on average, more than 500 tests were performed daily, of which 353 yielded positive results (0.18% of the total). Most of the screening involved rapid antigen tests (RAT), with 136,217 of these tests carried out, but 52,335 polymerase chain reaction (PCR) tests were also performed, particularly during the early stages of the epidemic. During this period, nine employees were responsible for contact-tracing and following up all positives. Besides this, after the detection of a positive case through the screening tests, the Catalan government was in charge of the trace and quarantine actions. A team of trackers monitored citizens through phone calls. The company experienced no outbreaks during the study period.

### 2.2. Model and Study Parameters

The model used here for measuring the impact of a test−trace−quarantine (TTQ) strategy on worker health and health resource use is based on González López-Valcárcel et al. [29]. The effectiveness of the strategy was measured by estimating the number of infections avoided in the population as a whole, based on a set of parameters (Table 1). Among the parameters that could be observed empirically were the number and rate of positive tests and the costs of hospitalization and intensive care unit (ICU) stays. Regarding the figures for the use of resources, the hospitalization, ICU admission, and mortality rates declared for the Catalan territory since 11 May 2020 were used (they therefore reflect a situation that does not include the high figures observed during the first wave of the pandemic in Spain) [31].

Assumption-based coefficients included the number and rate of positives among close contacts, the cost of monitoring COVID-19 cases that do not require hospitalization (10 telephone contacts with the primary care physician at a unit cost of €28 each), the effective reproductive number, the rate of people detected who could become infected after being detected, and the quarantine adherence rate (all based on the same reference study) [29]. We also assumed that one third of COVID-19 cases requiring hospitalization will suffer from a long-term health complication; the costs associated with these consequences are estimated assuming an annual incremental cost of €1000 for the remaining life expectancy of citizens suffering from long-term complications, discounted at a 3% rate. In relation to the transmission potential at the time of detection (number of potential iterations of the model), a central position was assumed within the epidemic curve (2.5 in a range between 0 and 5), slightly lower than the reference model (3, in a range between 0 and 6). Finally, using the same parameters as in González López-Valcárcel, the gains in Quality-Adjusted Years of Life (QALY) of avoided deaths and long-term morbidity avoided associated with the RAT testing strategy were measured. Productivity costs related to long-term mortality and morbidity were not taken into account, given that the average age of infected citizens who died or had moderate or severe symptoms in Spain is similar to or greater than the age of retirement [31]. Three sub analyses were carried out, one for each test type (PCR and RAT), as well as a calculation based on high incidence. Grouping was performed using the weighted average according to the relative weight of each type of test.

### 2.3. Testing

The detection of SARS-CoV-2 infection was performed using a rapid test for the qualitative detection of SARS-CoV-2 antigen Panbio™ COVID-19 Ag Rapid Test Device (Abbott) [32]. RNA extraction from Nasopharyngeal swabs was performed with the automated workstation KingFischer (ThermoFischer, Massachusetts, MA, USA), using a viral RNA/pathogen nucleic acid isolation kit (Thermofischer) following manufacturer’s instructions. RNA was subsequently PCR amplified using TaqMan 2019-nCoV Assay Kit (ThermoFischer) in the Applied Biosystems 7500 or QuantStudio5 Real-Time PCR instruments (ThermoFischer), following manufacturer’s protocol and recommendations. Positivity was considered when an amplification curve with a Ct < 37 was detected for two or more SARS-CoV-2 targets.

## 3. Results

Table 2 shows the results for the baseline scenario (all tests). The intervention identified a total of 353 positives, representing a total of 1082 avoided cases (3.07 avoided cases per positive). Among these, 34 hospitalizations, 2 ICU admissions, 11 cases with permanent sequelae, and 6 deaths were avoided; the rest (1029) were assumed to be cases treated at home. The set of avoided cases represents €744,488 of saving in the use of health resources (39%, 27%, 13%, and 21% of this value corresponds to COVID cases treated at home, hospitalizations, ICU admissions, and cases with permanent sequelae, respectively). If the 30.02 and 18.92 QALY relative to the morbidity and mortality avoided, respectively, are monetized, it is necessary to add €1,223,661 to the impacts in order to reflect improvements in the state of health of the population (61% corresponding to the morbidity avoided).

### Sensitivity Analysis

The same analysis was applied separately for the PCR and RAT screenings (Table 3). It should be noted that the difference in the results, relative to the base case, between these scenarios is attributable not only to the type of test, but also to the set of circumstances in which they were used (e.g., RAT tests became available only during the second wave). The higher rate of positive results from PCR testing explains the greater impact of this kind of test. However, the positive result rates from either RAT or PCR seen in the screening program were lower than the positivity rates observed in the screenings of asymptomatic populations carried out by the Catalan public health system (1.37%). Likewise, the rates of use of health resources (hospitalization and ICU admission) and health impacts (mortality and permanent sequelae) observed in Catalonia were approximately between 30% and 50% lower, respectively, than in the rest of Spain. For this reason, a fourth scenario was proposed using the same rates as seen in the rest of Spain. In this scenario, the impacts on health and healthcare resource savings could be as high as €130.24 social savings per person screened.

## 4. Discussion

Until the various COVID-19 vaccines became available, mass screening has largely been implemented to allow institutions such as hospitals to carry on with normal activities by detecting any presence of the virus in the hospital community and then arresting its spread [33]. However, the present analysis shows that the benefits of such screening strategies go well beyond the immediate institution where it is carried out by quantifying the significant impacts they have for society at large. It should be noted that, according to recent studies, in a work context, the number of close contacts for any given individual may be as high as seven (as opposed to the three assumed here) [24]; if this is so, the value of the results presented here would be even greater.

In comparison with the results shown by this research, the results of the positivity rate in population surveillance in the Catalan context (1.37%) [34] are much greater. This is due to the different intensity of the testing strategy and the population profile analyzed. Nevertheless, this suggests that these actions would be especially relevant for population bands especially susceptible to having the virus. The potential heterogeneity among the commercial products that these different testing strategies use must also be taken into account.

Our results have two important implications. On the one hand, given the important internal benefits in terms of protected employee health and consequent uninterrupted productivity, companies should implement systematic mass workplace COVID-19 screening programs. On the other hand, governments have a great deal to gain in terms of savings for public expenditure by encouraging such workplace screening in the private sector, and should therefore encourage and promote them by means of, for example, subsidies.

## 5. Conclusions

In the context of COVID-19, mass workplace testing strategies undertaken through private initiatives also confer considerable public benefits. For the case studied here of a large company in Catalonia, the financial impact of screening meant savings in social costs of €10.44 per test performed and €5575.59 per positive case detected; 38% of this figure corresponds to savings derived from health resources that were avoided (hospital beds, ICU admissions, and follow-up of positive cases treated at home), while the remaining 62% corresponds to savings derived from negative health impacts (morbidity and mortality avoided). In scenarios characterized by higher positivity rates and consequently greater impacts on health and health resource use, these effects could be up to ten times greater (€130.24 per test performed or €69,565.59 per positive detected).

## Figures and Tables

**Table 1 ijerph-18-07129-t001:** Base scenario settings (all screenings).

Parameter	Value in Base Scenario
Tests performed	188,552
Cost 10 follow-up calls to COVID-19 cases treated at home	€280
Cost of a COVID-19 hospitalization	€6050
Cost of admission to COVID-19 ICU	€43,400
Cost of permanent COVID-19 sequelae discounted at 3%	€14,754
Positivity rate	0.19%
Close contacts for COVID-19 case	3
% Positive among close contacts	41%
% Adherence to quarantine	75%
% Detected that could infect after detected	80%
Effective reproductive number	1
Number of iterations	2.5
Hospital admission rate	3.1%
ICU admission rate	0.2%
Lethality rate	0.6%
Permanent sequelae rate	1.0%
QALY lost due to sequelae discounted at 3%	2.78
QALY lost by mortality at 3%	2.92
Monetary value of a QALY	€25,000

**Table 2 ijerph-18-07129-t002:** Economic and health consequences of a mass workplace testing strategy in Catalonia.

	Quantity	Cost/Unit	Total
**Health resources use avoided**			
Total COVID-19 cases avoided	1082		
COVID-19 cases treated at home	1029	280	288,072
Hospitalizations	34	6050	202,899
Admissions to ICU	2	43,400	93,903
Cases with permanent sequelae	11	14,754	159,614
Total monetary savings due to health resource use avoided			€744,488
**Health impacts avoided**			
QALY gained by morbidity avoided			30.02
QALY gained by mortality avoided			18.92
Total monetary savings due to health impacts avoided			€1,223,661
**Total savings**			
Social savings per positive detected			€5,575.49
Social savings per test			€10.44

**Table 3 ijerph-18-07129-t003:** Summary of the social savings of the screening policy (€/test performed).

Stage	Description		Impact on Use of Health Resources (€)	Monetized Health Impact (€)	Total (€)
1	Base scenario		3.95	6.49	10.44
2	PCR	Positivity rate: 0.51% Positive among close contacts: 24%	8.38	13.77	22.16
3	RAT	Positivity rate: 0.06% Positive among close contacts: 48%	1.39	2.28	3.66
4	High incidence hypothesis	Positivity rate: 1.37% Hospitalization rate: 5.5% ICU rate: 0.4% Mortality rate: 0.9% Permanent sequelae rate: 2%	44.39	85.84	130.24

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
