# Peer review of "Impact of Mass Workplace COVID-19 Rapid Testing on Health and Healthcare Resource Savings"

_ijerph, 2021, doi:10.3390/ijerph18137129_

Round 1
Reviewer 1 Report
The problem with the article, as with most modelling studies, is that it relies on assumptions. In this case assumptions that come with substantial uncertainties. On is that the company was a front runner during the heigh days of the pandemic (meaning a positive bias on the results, that from my perspective are not that impressive). Others reflect the cost averages and of course the numbers of positive cases. The methodology is acceptable, but not novel in any way.
The authors quite strongly state that private companies have positive external effects as a result of mass testing. Governments might consider to subsidize them. Here I have hesitations. Yes, they might but companies - car companies for sure - also come with negative externalities, so governments might tax them. Depending on the specific activity all kinds of external effects can be calculated. Here the authors choose one specific case: mass testing. So we lack a broader perspective. Maybe the state has helped the company in many other ways and of course other external effects are also relevant.
I thus recommend to reject the paper, but if your editorial policies do not take my arguments as valid you might publish after some minor revisions (there are no serious methodological flaws), most notable a stronger discussion on limitations of the work.
Author Response
- Dear Reviewer 1,
Thank you very much for your contributions, which we respond to below.
"The problem with the article, as with most modelling studies, is that it relies on assumptions. In this case assumptions that come with substantial uncertainties. On is that the company was a front runner during the heigh days of the pandemic (meaning a positive bias on the results, that from my perspective are not that impressive). Others reflect the cost averages and of course the numbers of positive cases. The methodology is acceptable, but not novel in any way."
- Thank you so much for your comments. This article, as you rightly mention, uses a previously published methodology. Assumptions are part of it. Costs and number of positive cases, on the other hand, are observational data.
"The authors quite strongly state that private companies have positive external effects as a result of mass testing. Governments might consider to subsidize them. Here I have hesitations. Yes, they might but companies - car companies for sure - also come with negative externalities, so governments might tax them. Depending on the specific activity all kinds of external effects can be calculated. Here the authors choose one specific case: mass testing. So we lack a broader perspective. Maybe the state has helped the company in many other ways and of course other external effects are also relevant."
- The present article points out that screenings carried out by companies have positive social effects that can be recognized in different ways (subsidies are not the only one, also administrative and bureaucratic agility). It is clear that the negative social effects of industrial activity, known and studied, must be assessed as it is already being done now. This article does not mean otherwise.
"I thus recommend to reject the paper, but if your editorial policies do not take my arguments as valid you might publish after some minor revisions (there are no serious methodological flaws), most notable a stronger discussion on limitations of the work. "
- During this review we have made changes to the discussion. We believe that we have addressed the issues raised by the reviewers. We hope that the article is in good condition to be published in its current form.
On behalf of my coauthors, thank you for your help.
Reviewer 2 Report
This paper utilizes COVID-19 test numbers and positivity rates of a large automotive company as a case study to model the impact on public health and healthcare resource savings using cost analysis methods. While the findings of the paper have important implications for justifying the social value of population health initiatives in employee settings, I have a few comments which would improve the quality of the manuscript.
General:
Methodology: Please elaborate on details in the methodology. Please specify how and when individuals were tested. Was it based on symptom tracking? Was it population level testing at a given frequency? How many tests were performed per employee? At what frequency? Why were 500 tests per day performed? Was this population surveillance?
Methodology: Please describe the methods for detection and quarantine. Please describe the test-trace-quarantine (TTQ) strategy.
Methods: Please describe the symptom tracking
Methods: Please describe the tests used and test accuracy.
Methods: Please clarify. Was PCR testing used subsequent to RAT testing to confirm positive testing?
Methods:
Please address how the positivity numbers in the company compare to the population in the community at the similar time. (perhaps in the discussion) The numbers seem low. However, if they represent population surveillance, they may be appropriate.
Specific:
A few typos were detected. Please correct.
Methodology: Please change “The sample to be analyzed” to “The sample analyzed”
Results / line 2: Please change “identifies” to “identified”
Table 3 / Line 3: should this be RAT?
Discussion: Please change “one the one hand” to “On the one hand”
Ref 23. Please correct author spelling to Plantes
Author Response
Reviewer 2:
- Dear Reviewer 2,
Thank you very much for your contributions, which we respond to below.
"This paper utilizes COVID-19 test numbers and positivity rates of a large automotive company as a case study to model the impact on public health and healthcare resource savings using cost analysis methods. While the findings of the paper have important implications for justifying the social value of population health initiatives in employee settings, I have a few comments which would improve the quality of the manuscript."
General:
"Methodology: Please elaborate on details in the methodology. Please specify how and when individuals were tested. Was it based on symptom tracking? Was it population level testing at a given frequency? How many tests were performed per employee? At what frequency? Why were 500 tests per day performed? Was this population surveillance?"
- Thank you so much for your comments. All workers in the production chain were tested twice a week. We have added this piece of information in the "Methodology" section.
"Methodology: Please describe the methods for detection and quarantine. Please describe the test-trace-quarantine (TTQ) strategy. Please describe the symptom tracking. "
- After the detection of positive cases through the screening tests, the Catalan government was in charge of the trace and quarantine actions. A team of trackers monitored citizens through phone calls. This information has been added to the text.
"Methods: Please describe the tests used and test accuracy."
- Detection of SARS-CoV-2 infection was performed using a rapid test for qualitative detection of SARS-CoV-2 antigen Panbio™ COVID-19 Ag Rapid Test Device (Abbott) (J Infect. 2021 May;82(5):186-230. doi: 10.1016/j.jinf.2020.12.033). RNA extraction from Nasopharyngeal swabs was performed with the automated workstation KingFischer (ThermoFischer); using Viral RNA/Pathogen Nucleic Acid Isolation kit (Thermofischer) following manufacturer's instructions. RNA was subsequently PCR amplified using TaqMan 2019-nCoV Assay Kit (ThermoFischer) in the Applied Biosystems 7500 or QuantStudio5 Real-Time PCR instruments (ThermoFischer) following manufacturer’s protocol and recommendations. Positivity was considered when an amplification curve with a Ct<37 was detected for two or more SARS-CoV-2 targets.
This information has been added to the text. Please check the new version of the draft.
"Methods: Please clarify. Was PCR testing used subsequent to RAT testing to confirm positive testing?"
- This information has been added to the text. Thank you very much.
"Please address how the positivity numbers in the company compare to the population in the community at the similar time. (perhaps in the discussion) The numbers seem low. However, if they represent population surveillance, they may be appropriate."
- Thank you for your comment. Indeed, they represent population surveillance. The positivity rate (0,18%) is quite different compared to the the community testing figures (1.37%, as reported in other studies performed in Catalonia*), but this is due to multiple issues (the intensity of the testing strategy, the population profile analyzed...). This issue is mentioned in the discussion.
*López Seguí, F.; Estrada Cuxart, O.; Mitjà i Villar, O.; Hernández Guillamet, G.; Prat Gil, N.; Maria Bonet, J.; Isnard Blanchar, M.; Moreno Millan, N.; Blanco Guillermo, I.; Vilar Capella, M.; Català Sabaté, M.; Aran Solé, A.; Argimon Pallàs, J.M.; Clotet, B.; Ara del Rey, J. A Cost-Benefit Analysis of the COVID-19 Asymptomatic Mass Testing Strategy in the North Metropolitan Area of Barcelona. Preprints 2021, 2021050327 (doi: 10.20944/preprints202105.0327.v1).
Specific:
A few typos were detected. Please correct.
- Thank you very much. All typos have been corrected in the new version of the draft.
Reviewer 3 Report
Estimated Authors,
Estimated Editors,
I've read with great interest the present paper from the SEAT Occupational Medicine service of Barcelona. In fact, the potential significance of this research fairly exceeds the restricted field of the occupational medicine, with substantial overlapping with public health.
In my opinion, however, the paper - in the present stage, cannot be accepted for an eventual publication on IJERPH for the following reasons:
- The screening program must be more precisely detailed. We know only how many test were performed, but the sampling strategy is not reported. Did the Authors implement a periodic sampling strategy? or else?
- What about the RAT you actually employed? Available reports suggest a significant heterogeneity among the commercial products that ware made available up to the present day. Therefore the specific testing assays must be reported.
- The setting of the sampling strategy must be more properly described. Were the sampling performed by skilled medical of paramedical personnel? or did you refer to personnel from the occupational medicine service that received a focused retraining? As an incorrect sampling may impair the validity of the sampling, this factors should be at least reported and discussed.
- Please include in Table 1 the cost for RAT and PCR test
- Table 3 should include the Units of Measure, particularly in the last 3 columns (it is reported in the main caption, but please include it)
- Discussion should be partially revised in order to include some insights drawn from the sampling strategy and the potential difficulties in obtaining the adherence of workers to the sampling strategy you applied.
Author Response
"The screening program must be more precisely detailed. We know only how many test were performed, but the sampling strategy is not reported. Did the Authors implement a periodic sampling strategy? or else?"
- Dear reviewer, thank you very much for your comments. All workers in the production chain were tested twice a week. This was an important piece of information that was missing in the text and has now been added to the Methods section. Please see the new version of the draft.
"What about the RAT you actually employed? Available reports suggest a significant heterogeneity among the commercial products that ware made available up to the present day. Therefore the specific testing assays must be reported."
- The specific test used has been added to the text. Moreover, the heterogeneity that you mention has been added in the discussion as a limitation.
"The setting of the sampling strategy must be more properly described. Were the sampling performed by skilled medical of paramedical personnel? or did you refer to personnel from the occupational medicine service that received a focused retraining? // As an incorrect sampling may impair the validity of the sampling, this factors should be at least reported and discussed. // Discussion should be partially revised in order to include some insights drawn from the sampling strategy and the potential difficulties in obtaining the adherence of workers to the sampling strategy you applied".
- As mentioned, there was no sampling, as all workers in the production chain were tested twice a week. Sorry for the confusion.
"Please include in Table 1 the cost for RAT and PCR test"
- Dear reviewer. For privacy reasons, the promoter of the study does not disclose the costs it has incurred. The purpose of the study is to quantify the costs avoided in the public sector.
"Table 3 should include the Units of Measure, particularly in the last 3 columns (it is reported in the main caption, but please include it)."
- Done. Thank you very much for your suggestion.
Round 2
Reviewer 3 Report
Estimated Authors,
I've read with interest this revised version of your paper "Impact of mass workplace COVID-19 rapid testing on health and healthcare resource savings". The large majority of all my previous concerns were properly addressed - but one.
Unfortunately, Authors state that:
"For privacy reasons, the promoter of the study does not disclose the costs it has incurred. The purpose of the study is to quantify the costs avoided in the public sector".
Even though I clearly understand the rationale behind the choice of the promoter, Authors should be aware that such statement represents, from my point of view as a reviewer, a major and substantially critical issue for a paper that is centered on the savings from a mass testing campaign.
In your analyses, you've reported on the savings from the public point of view, but workplace screenings (from the general framework of occupational medicine in European Union) are usually paid by the Employer (there are obviously some exceptions, with costs paid by the Regional/National Health Authorities, e.g. Regional Health Authorities from Emilia Romagna in Italy). Therefore, it may be perceived as somewhat unethical report on 130€ of savings when we're unaware of the costs sustained by the employer. It may be perceived as a cost-shifting from public to private, and vice-versa in terms of benefits, and therefore the statement that "... governments have a great deal to gain in terms of sacing from the public expenditure by encouraging such workplace screening in the private sector..." is misleading.
Therefore, from my point of view, the present paper could be accepted immediately after the inclusion of such critical issue (i.e. the costs sustained by the parent company, with its clear and evident reporting), otherwise I'm definetely suggesting the reject of this paper.
Author Response
Dear Reviewer 3,
In the first review we already explained that including the costs incurred by the companies is out of the focus of the article, being the only point of interest the social impacts (externalities) of workplace testing strategies. Including further information would lead to a different type of economic analysis (a Cost-Benefit-Analysis), with a different methodology and focus. This is not the analysis we are proposing on this paper. Furthermore, we do not understand why the promoter of these testing interventions should make public the costs incurred, as it is in its right to do so or not.
We have already transmitted our position to the Academic Editor so that the article can be published in its current status.
Best regards,
The authors